# Quantifying the impacts of volume-based procurement policy on spatial accessibility of antidepressants via generic substitution: A four-city cohort study using drug sales data

**Aoming Xue[1‡], Qingyuan Xue[2‡], Jiahong Fu[3], Keye Fan[1], Jiale Zhang[4], Peiyan Cai[5], Yuanyuan Kuang[6], Yingsong Chen[7], Jifang Zhou[1]***, Bin Jiang[4,8]***

**1** School of International Business, China Pharmaceutical University, Jiangsu, China, **2** School of health management, Inner Mongolia Medical University, Inner Mongolia, Hohhot, China, **3** Department of Philosophy, Peking University, Beijing, China, **4** School of Pharmaceutical Sciences, Peking University Health Science Center, Beijing, China, **5** School of Pharmaceutical Sciences, China Medical University, Liaoning, Shenyang, China, **6** College of Pharmacy, Inner Mongolia Medical University, Hohhot, China, **7** National and Local Joint Engineering Research Center for Mongolian Medicine Research and Development, Tongliao, China, **8** Public Policy Research Center, Peking University, Beijing, China

‡ AX and QX contributed equally to this work and share first authorship.
* 1020202613@cpu.edu.cn (JZ); binjiang@hsc.pku.edu.cn (BJ)

**Data Availability Statement:** Data cannot be shared publicly because they are owned by the Chinese Pharmaceutical Association. Data are

## Abstract

### Objective

To assess the spatial accessibility and inequality of antidepressants and its correlation with VBP (Volume-based procurement) policy using procurement data from four representative Chinese cities between 2018 and 2020.

### Methods

The least-cost-path algorithm was employed to calculate travel time from each population point to the nearest medical institution. Gini coefficient and Theil index were utilized to measure accessibility and inequality. OLS (Ordinary Least Squares) and mediation analysis were used to investigate potential statistical relationships.

### Results

Under the influence of the VBP policy, we observed varying degrees of growth in the procurement volumes of two antidepressants across different cities (Escitalopram: Beijing 30.3%, Shanghai 26.2%, Ningbo 37.4%, Harbin 25.7%; Paroxetine: Beijing 28.2%, Shanghai 1.2%, Ningbo 50.2%, Harbin 590.5%). The increase in the procurement volumes of antidepressants across cities was primarily driven by generic drugs (Escitalopram: Beijing 159.8%, Shanghai 75.0%, Ningbo 146.4%, Harbin 146.3%; Paroxetine: Beijing 67.3%, Shanghai 4.9%, Ningbo 58.0%, Harbin 15,758.3%). In the results on spatial inequality, we observed annual improvements across all cities, with more pronounced progress in economically underdeveloped regions (Escitalopram: Gini in Harbin decreased by 10.6%; Paroxetine: Gini in Harbin decreased by 32.6%). In Beijing, the substitution of generic

available from the Chinese Pharmaceutical Association Science and Technology Development Center (contact via Xu Yingfeng, +8610-65662756) for researchers who meet the criteria for access to confidential data.

**Funding:** "National Natural Science Foundation of China: 72064032: Q.X. National Natural Science Foundation of China: 72274008: B.J. Natural Science Foundation of Beijing Municipality: 9232010: B.J. Natural Science Foundation of Inner Mongolia Autonomous Region: 2024MS07007: Q. X."

**Competing interests:** The authors have declared that no competing interests exist.

**Abbreviations:** VBP, Volume-based procurement; DDD, Defined daily dose; CNY, Chinese Yuan; OLS, Ordinary least squares; ACME, : The average causal mediation effects; ADE, Average direct effects.

escitalopram was found to be a partial mediating factor in the improvement of spatial inequality (ACME = -0.00, p-value = 0.01; ADE = -0.00, p-value = 0.02). In Harbin, the substitution of generic paroxetine was identified as a complete mediating factor for spatial inequality (ACME = -0.04, p-value = 0.01; ADE = 0.01, p-value = 0.14).

## Conclusions

This study found that the spatial accessibility and inequality of antidepressant medications gradually improved under the influence of the VBP policy. These improvements can be partially attributed to the substitution of generic drugs.

## Introduction

Depressive disorders are a group of common mental disorders which are the second most prevalent disorder globally after cardiovascular disease. In China, depressive disorders have been estimated to be the second leading cause of years lived with disability, which means that depression has gradually become an important public health problem affecting the Chinese population [1]. It is estimated that the prevalence of depression in China rose from 3224.6/100,000 to 3990.5/100,000 from 1990 to 2017 [1]. In 2017, there were 56.36 million depression patients in China, accounting for 21.3% of the global cases.

However, because nationally representative epidemiological data for depressive disorders are unavailable in China, existing data of disease burden of depression, such as number of people with depression, affordability of drug costs, can only be estimated from mathematical models with broad uncertainty ranges [2], making evidence-based and precision mental health policy development difficult.

China is a mental health resource-poor country and mental health services are mainly concentrated in large psychiatric specialty hospitals in large cities in eastern and central China [3], resulting in geographical and rural–urban inequities in mental health services. People with depressive disorders receive inadequate treatment in many countries, particularly in low-income and middle-income countries, including China [4].

The disruptions to the health system caused by the COVID-19 pandemic further exacerbated the depressive management [5], with the following implications:(1) The rapid spread and heightened mortality risk associated with COVID-19 infection may elevate the likelihood of depressive disorders in healthy individuals within the general population. Additionally, for individuals already grappling with preexisting depressive conditions, the impact may exacerbate their depressive symptoms.(2) The COVID-19 pandemic has affected the provision of psychiatric care across the world [6]. The difficulty in maintenance treatment of individual patients with mental disorders is increased for several reasons. First, under strict control due to the epidemic, most of the general hospitals in China and elsewhere are temporarily unable to care for psychiatric patients to lower the possibility of COVID-19 transmission. Second, hospitalized patients and outpatients, requiring treatment or follow-up, have faced disruptions as a consequence of suspended public transport in various regions. This has accentuated the imbalance in mental health services and underscored spatial accessibility inequalities in China.

In order to reduce the medication burden of people and address the problems of difficult and expensive medical treatment, inconsistent quality of generic medicines, irregularities in the circulation of medicines, the geographical and rural-urban inequities, China began to implement the VBP (Volume-based procurement) in December 2018, starting with pilot

projects in the "4+7" cities and then expanding to the whole country [7]. The VBP policy follows the principle of "reducing prices through bulk purchasing," offering hospitals and patients more cost-effective medications. Nine rounds of procurement have been carried out so far, including 374 kinds of medicines with an average price reduction of more than 50%. In terms of the implementation effect of VBP, after eight rounds of procurement, the total procurement amount dropped from 348.66 billion yuan in 2019 to 186.22 billion yuan in 2023, showing a year-on-year decrease. The consumption volume, on the other hand, surged from 68.1 billion tablets/branches to 81.5 billion tablets/branches, increasing year by year. It has been four years since the first round of, and the actual supply of the bid-winning drugs exceeded more than two times the agreed annual procurement volume.

Depression medications have always been a crucial concern of the pooled procurement policy. In order to reduce the medication burden of depression patients and enhance the accessibility of drugs, nine types of depression medications have been included in the first eight rounds of VBP (S1 Table).

The VBP policy has been in operation for five years. While there is existing evidence on the effectiveness of it, limited researches specifically assessed whether the VBP policy addressed the problem of geographic disparities and enhanced the spatial accessibility of antidepressants. Studies on policy have mainly focused on non-spatial access to and affordability of bid-winning drugs [8]. Studies also have shown geographic disparities in drug supply within China [9]. Therefore, evaluation of the VBP is crucial to enhance the spatial accessibility of commonly used antidepressants from theoretical and practical aspects.

The overarching goal of this study is to evaluate the spatial accessibility of the first-round procured antidepressants in four Chinese typical cities (Beijing, Shanghai, Ningbo, and Harbin) from 2018 to 2020, by incorporating big data and scientific methods into the evaluation of inequalities in antidepressants spatial accessibility, and promoting further improvement of the pooled procurement policy. The specific aims of this study were threefold: 1) to estimate the price and volume of the first-round procured antidepressants in four cities. 2) to evaluate the spatial accessibility of essential anti-depressive agents in four cities. 3) to examine inequality of spatial accessibility to antidepressants using the Gini coefficient and Theil index among the four cities. 4) to estimate the mediation effects of Generic Drug Substitution on Unequal Spatial accessibility. This is the first study to quantitatively describe the status quo of spatial accessibility of the VBP drugs in China. High-resolution digital map of spatial accessibility of the VBP drugs from this study is intended to launch a new chapter in the quantitative assessment of the spatial accessibility of medicines in China, promoting long-term commitment to improving accessibility and affordability of national reimbursement drugs among China's residents. Meanwhile, to the extent that the COVID-19 pandemic actually affected spatial accessibility of the VBP drugs, this study possesses a significant benchmark record of drug procurement information from the pre-COVID-19 era, providing a reliable basis for accurate analysis. Furthermore, it extends its value into the COVID-19 era, offering a robust dataset that allows for precise and insightful analysis during this distinct period.

## Methods

### Study area

Two groups of sample cities were selected for this study, the first group being Beijing and Shanghai, and the second being Ningbo and Harbin. The reasons for choosing the above two groups of cities are as follows: (1) In the first group, Beijing is the capital city located in northern China, and Shanghai is the economic capital in the eastern region. The two mega-cities have large numbers of people with depressive disorders, for a large population and work

pressure, large mobile populations, and serious aging problems (especially in Shanghai). In addition, the two cities are mental health resource-rich cities where patients have high spatial accessibility. However, the higher income level of patients contributes to their strong demand for brand-name drugs. It is a very meaningful issue whether the bid-winning drugs in the VBP will change the medication habits of patients in the first group of cities from brand-name drugs to generic drugs and enhance the spatial accessibility of patients around the cities. (2) In the second group, Ningbo is a middle-sized city in the eastern part, and Harbin is situated in the northeastern region. The two cities are relatively close to each other in terms of population size and level of economic development. Mental health service inequality is a prominent problem between urban and rural areas in both cities, particularly in Harbin where the net population outflow phenomenon is more serious, with a large number of empty-nesters and obvious lack of health service capacity in the countryside. How the VBP policy affects depression patients' access to health care in such medium-sized cities is also an issue of concern. (3) It should be noted that because the level of health insurance digitalization is relatively weak in central and western cities of China, the quality and availability of data are not high enough to meet the requirements of the study. Hence, it is not considered in this study for the time being.

In this study, we selected two depression medicines of the first round of the VBP, paroxetine and escitalopram. The first round of the VBP has been implemented for a long time, making it easy to observe the change of the VBP policy on the medications of depression patients as well as spatial accessibility. Paroxetine and escitalopram are the two most frequently used antidepressants with a frequency of 18.8% and 17.8% respectively [10], which is highly representative (S1 Table).

## Data sources

Data for the study were obtained from multiple sources. Drug procurement information was collected from the China Pharmaceutical Association, which provides comprehensive data from 2018 to 2020 on national drug procurement (S2 Table). Population information was extracted from the websites of the National Statistical Bureau (https://www.stats.gov.cn). Geographic information was sourced from Gaode Location Service and Road Traffic Network (https://www.gaode.com). Additionally, population count data for 2020 was obtained from WorldPop (https://hub.worldpop.org) [11], and friction surfaces for calculating travel time were downloaded from the Malaria Atlas Project (https://malariaatlas.org) [12].

## Spatial accessibility

A grid with a resolution of 1x1 km$^2$ was overlaid onto the four cities [13]. The driving travel time from each grid cell to the nearest medical institution that procured antidepressants was used as an indicator of spatial accessibility [14]. The least-cost-path algorithm was employed to calculate travel time based on the friction surfaces [15]. Grid cells were categorized into different travel time intervals, and the analysis of travel time proportions was conducted to describe spatial accessibility [16].

## Measures of inequality

The Gini coefficient was used to measure the inequality of spatial accessibility [17]. A gravity decay model was employed to process travel time [18, 19], and the Gini coefficient was calculated based on the formula: $Gini = \sum_{k=1}^{N-1} (F_k \Phi_{k+1} - F_{k+1} \Phi_k)$. Here, N represents the total number of pixels across Nepal, and k denotes the rank based on the impedance function value, from the lowest to the highest. $F_k = \sum_{s=1}^{k} (p_s/P)$ and $\Phi_k = \sum_{s=1}^{k} (h_s/H)$, represented the

cumulative proportion of the population and the cumulative value of the impedance function up to k, respectively [13]. The Theil L index was also calculated to measure the level of inequality in spatial accessibility [20], the formula is as follows: $L = \sum_k (p_k/P)\log(X/x_k)$, where $X =$ H/P. Both within-group (city-level) and between-group Theil L indices were computed.

## Statistical analysis

Descriptive statistics were used to assess city-level characteristics of spatial accessibility and psychotropic medication procurement patterns. OLS (Ordinary Least Squares) regression models with robust standard errors were employed to evaluate the association between the VBP policy and travel time inequity. The analysis controlled for city-level sociodemographic characteristics and year of procurement.

## Mediation analysis

A mediation analysis was conducted to test the potential mediating effect of the VBP policy on the relationship between the Gini coefficient and each city [21]. The proportion of generic drugs was selected as the mediating variable. Other time-fixed variables and local policy effects were included in the mediation models as confounding factors. The ACME (average causal mediation effects) and ADE (average direct effects) were calculated to measure the strength of the mediating variable's effect.

## Visualization and statistical software

ArcGIS V.10.8 was used for data visualization, and R version 4.1.0 (R Foundation for Statistical Computing, Vienna, Austria) was used for statistical analyses.

## Statistical significance

A two-sided p-value less than 0.05 was considered statistically significant in the study.

## Results

### Changes in sales and prices

During the period from 2018 to 2020, the utilization of brand-name drugs of Escitalopram and Paroxetine increased (Table 1). A gradual rise of 29.2% was seen in Beijing, while in Ningbo, a more significant growth of 43.8% was demonstrated. The utilization of generic drugs also showed substantial growth, with Beijing and Ningbo witnessing 68.5% and 65.0% of increase, respectively. In contrast, Shanghai and Harbin had moderate increases or even declines in the utilization of brand-name drugs as well as generic drugs.

The medication prices of Escitalopram and Paroxetine also exhibited a trend of decline over the years (Table 1). In both Beijing and Shanghai, noticeable decreases were shown in the volume-weighted unit prices of Escitalopram and Paroxetine. Compared to brand-name drugs, generic drugs showed a more pronounced decline in the medication prices. In Ningbo, the medication prices remained relatively stable from 2018 to 2019, whereas significantly decreased from 2019 to 2020. Similar trend was shown in Harbin in the same period, particularly for generic drugs.

### Spatial accessibility

The spatial accessibility of essential anti-depressive agents was assessed in four cities (Table 2). At the city level, Beijing and Shanghai had relatively high spatial accessibility, with

**Table 1. Yearly changes in usage, weighted average price, and sales.**

**Escitalopram**

| | DDDs | | | Price (CNY per DDD) | | | Sales (CNY) | | |
|---|---|---|---|---|---|---|---|---|---|
| **Overall** | **2018** | **2019** | **2020** | **2018** | **2019** | **2020** | **2018** | **2019** | **2020** |
| Beijing | 12972094 | 16151902 | 16897237 | 10.97 | 8.26 | 7.25 | 142321406.60 | 133426397.40 | 122491098.00 |
| Shanghai | 6901738 | 7927181 | 8711704 | 10.72 | 7.23 | 6.04 | 73966298.58 | 57299398.54 | 52593070.27 |
| Ningbo | 1616840 | 1927469 | 2222267 | 9.80 | 9.42 | 5.71 | 15849573.78 | 18147206.84 | 12681080.64 |
| Harbin | 531076 | 757092 | 667664 | 12.27 | 11.67 | 7.00 | 6515307.22 | 8836105.20 | 4672715.25 |
| **Brand name** | **2018** | **2019** | **2020** | **2018** | **2019** | **2020** | **2018** | **2019** | **2020** |
| Beijing | 6494376 | 6135570 | 5896513 | 13.65 | 12.07 | 11.25 | 88637686.32 | 74057689.54 | 66342412.87 |
| Shanghai | 2980103 | 2094666 | 1846887 | 13.82 | 12.28 | 11.27 | 41194462.05 | 25716936.86 | 20809466.88 |
| Ningbo | 529214 | 510846 | 350721 | 13.81 | 13.81 | 12.39 | 7306933.30 | 7053323.70 | 4344138.43 |
| Harbin | 367143 | 483210 | 263683 | 13.70 | 13.49 | 11.59 | 5028325.52 | 6516827.60 | 3056916.25 |
| **Generics** | **2018** | **2019** | **2020** | **2018** | **2019** | **2020** | **2018** | **2019** | **2020** |
| Beijing | 6477718 | 10016332 | 11000724 | 8.29 | 5.93 | 5.10 | 53683720.25 | 59368707.85 | 56148685.15 |
| Shanghai | 3921635 | 5832515 | 6864817 | 8.36 | 5.41 | 4.63 | 32771836.53 | 31582461.68 | 31783603.39 |
| Ningbo | 1087626 | 1416623 | 1871546 | 7.85 | 7.83 | 4.45 | 8542640.48 | 11093883.14 | 8336942.21 |
| Harbin | 163933 | 273882 | 403981 | 9.07 | 8.47 | 4.00 | 1486981.70 | 2319277.60 | 1615799.00 |

**Paroxetine**

| | DDDs | | | Price (CNY per DDD) | | | Sales (CNY) | | |
|---|---|---|---|---|---|---|---|---|---|
| **Overall** | **2018** | **2019** | **2020** | **2018** | **2019** | **2020** | **2018** | **2019** | **2020** |
| Beijing | 8776365 | 10138340 | 11252488 | 5.28 | 3.50 | 2.43 | 46424908.93 | 35515503.87 | 27444526.08 |
| Shanghai | 11744315 | 12505055 | 11883824 | 5.81 | 3.16 | 2.12 | 68345977.55 | 39619975.24 | 25225564.50 |
| Ningbo | 2235048 | 2515153 | 3357524 | 4.90 | 5.52 | 2.26 | 10963629.28 | 13908431.57 | 7597744.80 |
| Harbin | 168733 | 357562 | 1165181 | 8.90 | 6.95 | 2.40 | 1502137.70 | 2488098.72 | 2802509.80 |
| **Brand name** | **2018** | **2019** | **2020** | **2018** | **2019** | **2020** | **2018** | **2019** | **2020** |
| Beijing | 3270195 | 2894818 | 2038386 | 8.42 | 7.17 | 6.00 | 27561181.74 | 20771835.51 | 12235259.70 |
| Shanghai | 2146651 | 1731325 | 1815752 | 8.47 | 6.72 | 5.01 | 18193598.08 | 11641545.08 | 9099073.83 |
| Ningbo | 400490 | 465595 | 458610 | 8.26 | 7.17 | 4.91 | 3312021.33 | 3340920.54 | 2254991.31 |
| Harbin | 80817 | 154607 | 165890 | 7.84 | 6.89 | 3.09 | 634077.91 | 1065403.59 | 513269.55 |
| **Generics** | **2018** | **2019** | **2020** | **2018** | **2019** | **2020** | **2018** | **2019** | **2020** |
| Beijing | 5506170 | 7243522 | 9214102 | 3.42 | 2.03 | 1.65 | 18863727.19 | 14743668.36 | 15209266.38 |
| Shanghai | 9597664 | 10773730 | 10068072 | 5.22 | 2.59 | 1.60 | 50152379.47 | 27978430.16 | 16126490.67 |
| Ningbo | 1834558 | 2049558 | 2898914 | 4.17 | 5.15 | 1.84 | 7651607.95 | 10567511.03 | 5342753.49 |
| Harbin | 4800 | 83680 | 761200 | 3.15 | 2.01 | 1.55 | 15156.00 | 168821.12 | 1186710.80 |

[a] DDDs = Total procurement quantity/DDD. The DDD for Escitalopram is 10 mg, and for Paroxetine, it is 20 mg.

[b] The calculation of price is based on the annual average price (CNY) per DDD.

approximately 80% to 90% of the population having travel times within 5 minutes for escitalopram and paroxetine. The population with travel times exceeding 90 minutes was negligible in these cities. In Ningbo, most of the population had travel times ranging from 0 to 15 minutes for both drugs, and the accessibility of generic drugs was higher than that of brand-name drugs. Harbin had a lower proportion of the population with travel times within 5 minutes for both drugs (escitalopram: 35.8%; paroxetine: 33.79%). In Harbin, approximately 10% to 20% of the population had travel times exceeding 90 minutes for these medications. Overall, there were more inequalities in Harbin regarding the distance to the nearest medical institutions, which means a longer distance to the nearest medical institutions in Harbin comparing to the other cities (Fig 1). In contrast, there were relatively less inequalities among provinces in Shanghai, Beijing and Ningbo, with a shorter distance to medical institutions (Figs 2–4).

**Table 2. Summaries of population proportion with different spatial accessibilities (%).**

| Motorized mode | Percentage in different travel time (%) | | | | | | |
|---|---|---|---|---|---|---|---|
| **Escitalopram** | | | | | | | |
| **Overall** | <5 mins | 5–15 mins | 15–30 mins | 30–60 mins | 60–90 mins | 90–120 mins | >120 mins |
| Beijing | 88.68 | 9.31 | 1.47 | 0.41 | 0.08 | 0.02 | 0.01 |
| Shanghai | 82.63 | 15.90 | 1.23 | 0.24 | 0.00 | - | - |
| Ningbo | 65.86 | 25.56 | 7.38 | 1.11 | 0.07 | 0.00 | 0.00 |
| Harbin | 35.85 | 15.22 | 13.20 | 15.03 | 9.18 | 5.95 | 5.57 |
| **Brand name** | <5 mins | 5–15 mins | 15–30 mins | 30–60 mins | 60–90 mins | 90–120 mins | >120 mins |
| Beijing | 81.96 | 14.37 | 2.99 | 0.53 | 0.10 | 0.03 | 0.01 |
| Shanghai | 74.66 | 22.42 | 2.57 | 0.35 | 0.00 | - | - |
| Ningbo | 58.81 | 29.90 | 9.84 | 1.36 | 0.08 | 0.01 | 0.00 |
| Harbin | 32.05 | 15.63 | 13.72 | 15.92 | 10.37 | 6.36 | 5.96 |
| **Generics** | <5 mins | 5–15 mins | 15–30 mins | 30–60 mins | 60–90 mins | 90–120 mins | >120 mins |
| Beijing | 87.59 | 10.26 | 1.61 | 0.42 | 0.08 | 0.02 | 0.01 |
| Shanghai | 82.58 | 15.93 | 1.24 | 0.24 | 0.00 | - | - |
| Ningbo | 57.99 | 30.83 | 9.69 | 1.40 | 0.08 | 0.00 | 0.00 |
| Harbin | 33.90 | 15.43 | 12.29 | 12.14 | 11.11 | 7.51 | 7.61 |
| **Paroxetine** | | | | | | | |
| **Overall** | <5 mins | 5–15 mins | 15–30 mins | 30–60 mins | 60–90 mins | 90–120 mins | >120 mins |
| Beijing | 89.84 | 8.31 | 1.43 | 0.33 | 0.07 | 0.02 | 0.01 |
| Shanghai | 91.71 | 7.25 | 0.94 | 0.11 | - | - | - |
| Ningbo | 66.28 | 24.80 | 7.75 | 1.09 | 0.07 | 0.01 | 0.00 |
| Harbin | 33.79 | 13.85 | 14.63 | 20.38 | 8.24 | 4.12 | 4.98 |
| **Brand name** | <5 mins | 5–15 mins | 15–30 mins | 30–60 mins | 60–90 mins | 90–120 mins | >120 mins |
| Beijing | 82.92 | 12.32 | 3.85 | 0.74 | 0.12 | 0.03 | 0.02 |
| Shanghai | 80.39 | 16.54 | 2.71 | 0.36 | 0.00 | - | - |
| Ningbo | 52.11 | 31.06 | 13.73 | 2.97 | 0.12 | 0.01 | 0.00 |
| Harbin | 29.59 | 12.31 | 10.19 | 14.67 | 14.16 | 9.45 | 9.63 |
| **Generics** | <5 mins | 5–15 mins | 15–30 mins | 30–60 mins | 60–90 mins | 90–120 mins | >120 mins |
| Beijing | 89.45 | 8.61 | 1.40 | 0.43 | 0.08 | 0.02 | 0.01 |
| Shanghai | 91.52 | 7.39 | 0.98 | 0.11 | - | - | - |
| Ningbo | 65.36 | 25.53 | 7.93 | 1.10 | 0.07 | 0.01 | 0.00 |
| Harbin | 30.96 | 15.86 | 13.66 | 19.02 | 8.14 | 4.94 | 7.42 |

[a] The travel time (mins) proportion of population points within city to the nearest mental health facility.

## Inequality

For a further study of the changes in the inequality of spatial accessibility with regard to selected drugs in four cities over three years, the Gini coefficient, which is a measurement of inequality was applied [22]. The study revealed significant variations in the inequity of spatial accessibility to antidepressants among the four cities (Table 3). Beijing showed overall inconspicuous changes in the Gini coefficient. Shanghai witnessed a slight increase in the Gini coefficient for brand-name drugs but a decrease for generic drugs. In Ningbo, both brand-name and generic drugs showed a decrease in the Gini coefficient. Harbin exhibited a noticeable decrease in the Gini coefficient for both drugs, indicating a more equitable distribution of the drug supply, especially for generic paroxetine.

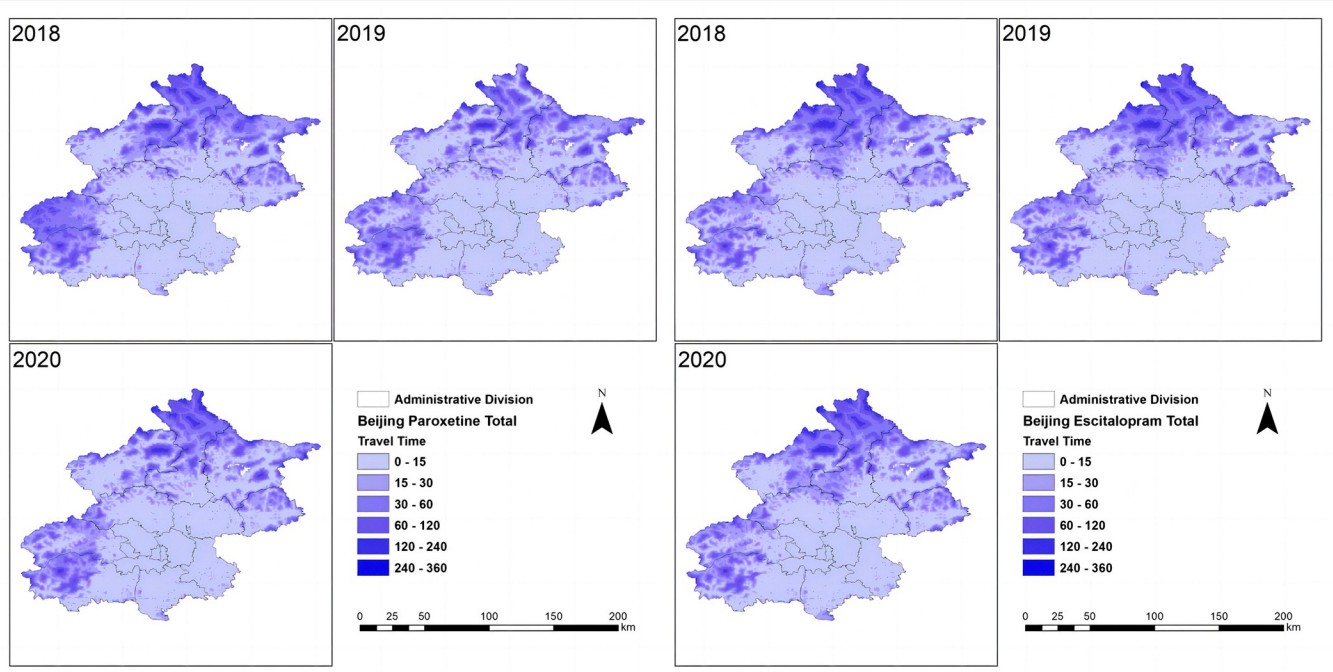

**Fig 1. Beijing spatial accessibility (mins) in total.** The shortest travel time from each 1 km² population point to the nearest mental healthcare of two drugs in Beijing was categorized into the following intervals: 0–15, 15–30, 30–60, 60–120, 120–240, and 240–360 minutes.

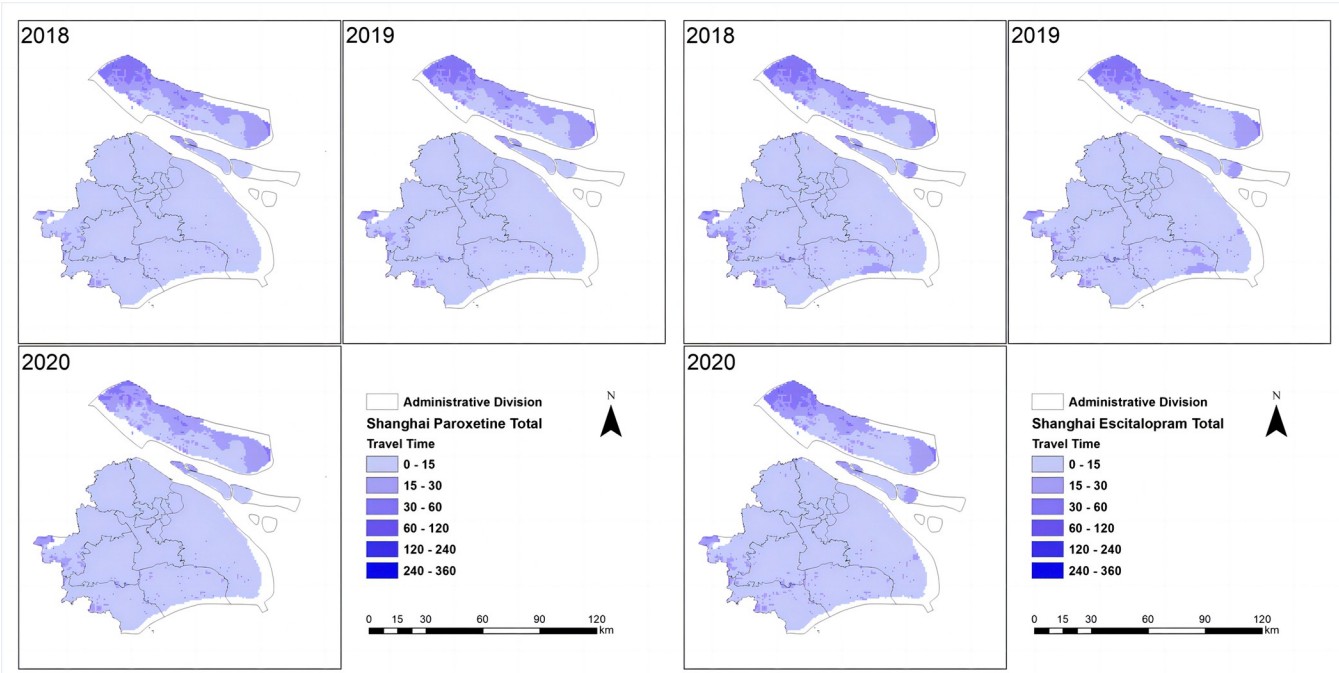

**Fig 2. Shanghai spatial accessibility (mins) in total.** The shortest travel time from each 1 km² population point to the nearest mental healthcare of two drugs in Shanghai was categorized into the following intervals: 0–15, 15–30, 30–60, 60–120, 120–240, and 240–360 minutes.

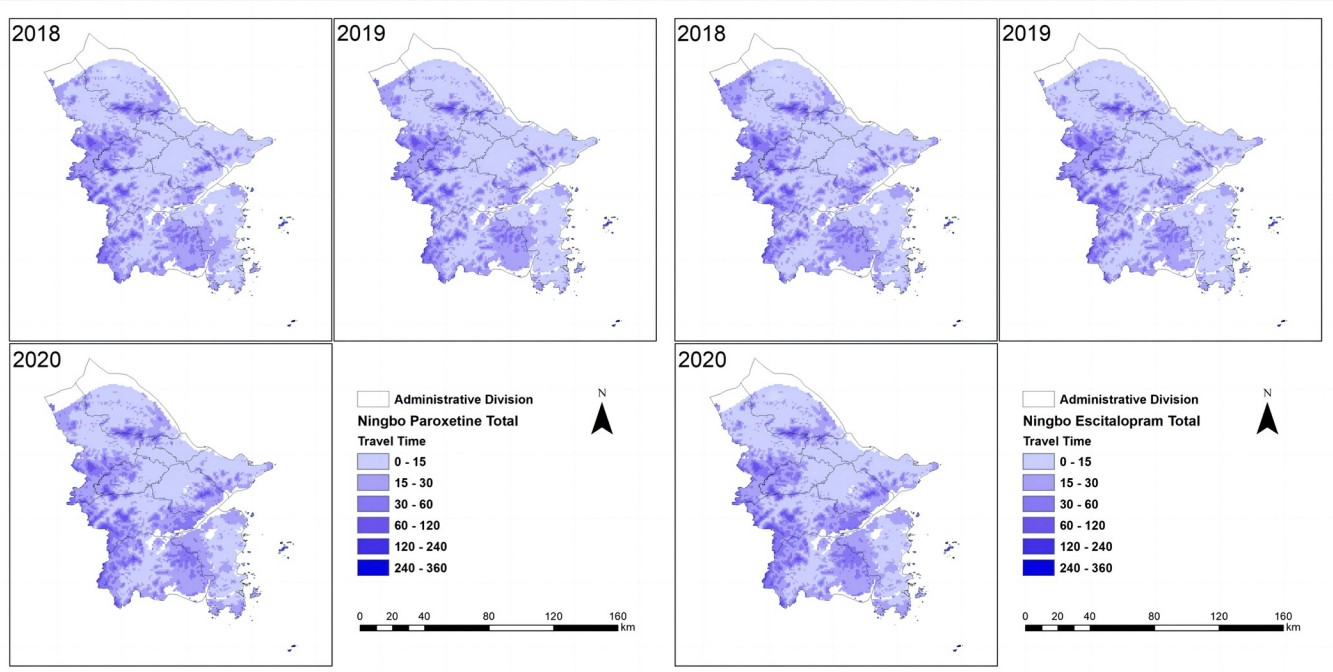

**Fig 3. Ningbo spatial accessibility (mins) in total.** The shortest travel time from each 1 km² population point to the nearest mental healthcare of two drugs in Ningbo was categorized into the following intervals: 0–15, 15–30, 30–60, 60–120, 120–240, and 240–360 minutes.

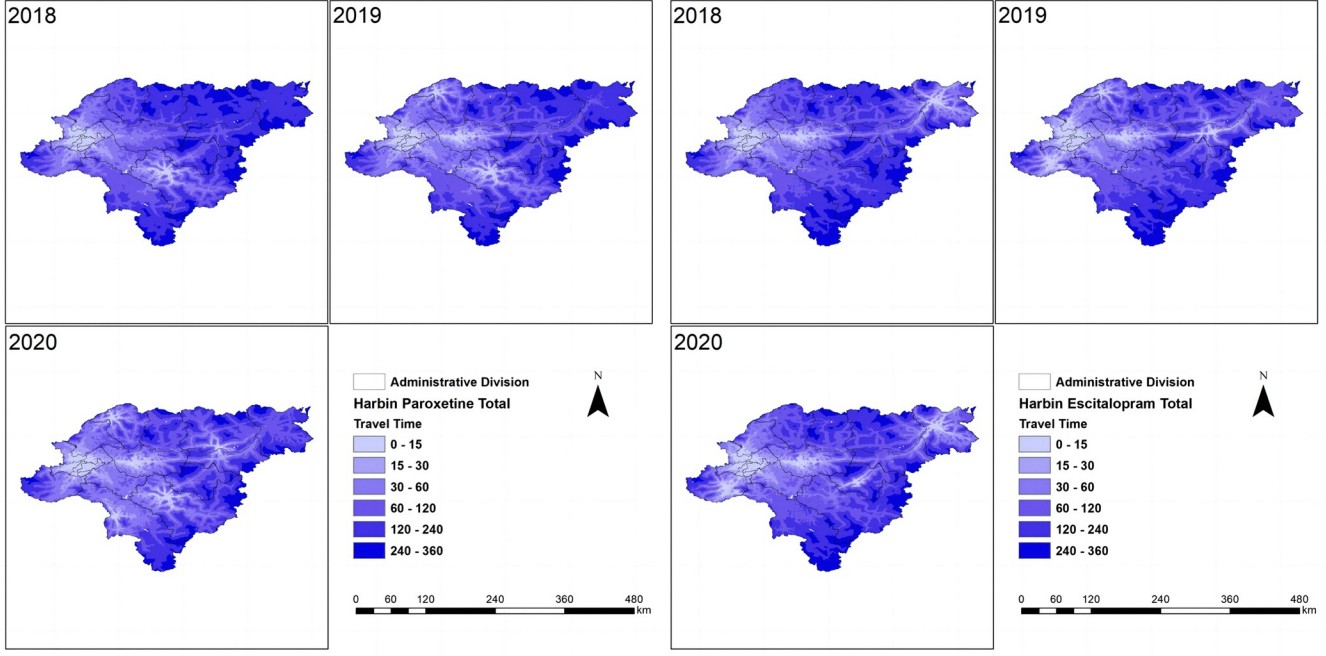

**Fig 4. Harbin spatial accessibility (mins) in total.** The shortest travel time from each 1 km² population point to the nearest mental healthcare of two drugs in Beijing was categorized into the following intervals: 0–15, 15–30, 30–60, 60–120, 120–240, and 240–360 minutes.

**Table 3. Inequality of spatial accessibility and the decomposition.**

**Motorized mode**

**Escitalopram**

| | Gini | | | Theil L | | | | | | | | |
|---|---|---|---|---|---|---|---|---|---|---|---|---|
| | | | | Theil L | | | Within-city (% of overall) | | | Between-city (% of overall) | | |
| Overall | 2018 | 2019 | 2020 | 2018 | 2019 | 2020 | 2018 | 2019 | 2020 | 2018 | 2019 | 2020 |
| Beijing | 0.00809 | 0.00737 | 0.00731 | 0.00311 | 0.00304 | 0.00284 | 96.64894 | 96.69689 | 96.74121 | 3.35106 | 3.30311 | 3.25879 |
| Shanghai | 0.00686 | 0.00631 | 0.00601 | 0.00046 | 0.00043 | 0.00042 | 80.29508 | 81.13784 | 81.51594 | 19.70492 | 18.86216 | 18.48406 |
| Ningbo | 0.02431 | 0.01830 | 0.02597 | 0.00352 | 0.00275 | 0.00398 | 92.35013 | 96.01755 | 95.07444 | 7.64987 | 3.98245 | 4.92556 |
| Harbin | 0.33407 | 0.29441 | 0.29871 | 1.04779 | 0.82733 | 0.82514 | 69.96108 | 73.98329 | 74.83240 | 30.03892 | 26.01671 | 25.16760 |
| **Brand name** | 2018 | 2019 | 2020 | 2018 | 2019 | 2020 | 2018 | 2019 | 2020 | 2018 | 2019 | 2020 |
| Beijing | 0.01292 | 0.01338 | 0.01398 | 0.00400 | 0.00410 | 0.00441 | 87.50930 | 84.88366 | 86.59876 | 12.49070 | 15.11634 | 13.40124 |
| Shanghai | 0.00902 | 0.01801 | 0.02779 | 0.00063 | 0.00682 | 0.01115 | 71.57295 | 65.43715 | 50.95871 | 28.42705 | 34.56285 | 49.04129 |
| Ningbo | 0.03136 | 0.02224 | 0.02921 | 0.00482 | 0.00327 | 0.00467 | 85.59673 | 94.08168 | 94.60302 | 14.40327 | 5.91832 | 5.39698 |
| Harbin | 0.33450 | 0.36010 | 0.36384 | 1.05043 | 1.42486 | 1.17246 | 69.91223 | 55.71284 | 65.33892 | 30.08777 | 44.28716 | 34.66108 |
| **Generics** | 2018 | 2019 | 2020 | 2018 | 2019 | 2020 | 2018 | 2019 | 2020 | 2018 | 2019 | 2020 |
| Beijing | 0.01042 | 0.00771 | 0.00753 | 0.00366 | 0.00306 | 0.00286 | 95.70782 | 96.59337 | 96.66146 | 4.29218 | 3.40663 | 3.33854 |
| Shanghai | 0.00909 | 0.00631 | 0.00601 | 0.00062 | 0.00043 | 0.00042 | 70.27273 | 81.14196 | 81.51594 | 29.72727 | 18.85804 | 18.48406 |
| Ningbo | 0.03115 | 0.02281 | 0.02653 | 0.00426 | 0.00333 | 0.00405 | 91.13788 | 96.29327 | 95.32801 | 8.86212 | 3.70673 | 4.67199 |
| Harbin | 0.38017 | 0.33727 | 0.31977 | 1.31528 | 1.02456 | 0.99651 | 63.51906 | 71.27423 | 69.49401 | 36.48094 | 28.72577 | 30.50599 |
| **Paroxetine** | | | | | | | | | | | | |
| **Overall** | 2018 | 2019 | 2020 | 2018 | 2019 | 2020 | 2018 | 2019 | 2020 | 2018 | 2019 | 2020 |
| Beijing | 0.00800 | 0.00637 | 0.00659 | 0.00326 | 0.00219 | 0.00256 | 96.65108 | 96.99969 | 97.36663 | 3.34892 | 3.00031 | 2.63337 |
| Shanghai | 0.00434 | 0.00416 | 0.00378 | 0.00037 | 0.00038 | 0.00022 | 78.48697 | 79.23150 | 77.80257 | 21.51303 | 20.76850 | 22.19743 |
| Ningbo | 0.02162 | 0.01979 | 0.02574 | 0.00313 | 0.00293 | 0.00389 | 94.68779 | 94.13681 | 94.58642 | 5.31221 | 5.86319 | 5.41358 |
| Harbin | 0.35900 | 0.30607 | 0.24196 | 1.36515 | 0.91587 | 0.60943 | 57.83490 | 69.07068 | 82.72514 | 42.16510 | 30.92932 | 17.27486 |
| **Brand name** | 2018 | 2019 | 2020 | 2018 | 2019 | 2020 | 2018 | 2019 | 2020 | 2018 | 2019 | 2020 |
| Beijing | 0.01319 | 0.01437 | 0.01416 | 0.00502 | 0.00465 | 0.00447 | 93.80988 | 91.51730 | 92.09987 | 6.19012 | 8.48270 | 7.90013 |
| Shanghai | 0.00856 | 0.00876 | 0.01267 | 0.00064 | 0.00067 | 0.00103 | 65.46383 | 65.84042 | 55.25105 | 34.53617 | 34.15958 | 44.74895 |
| Ningbo | 0.04329 | 0.03325 | 0.03498 | 0.00941 | 0.00564 | 0.00582 | 73.91922 | 88.90861 | 90.61215 | 26.08078 | 11.09139 | 9.38785 |
| Harbin | 0.39931 | 0.39923 | 0.36167 | 1.64144 | 1.64151 | 1.10816 | 56.03204 | 56.02234 | 69.49168 | 43.96796 | 43.97766 | 30.50832 |
| **Generics** | 2018 | 2019 | 2020 | 2018 | 2019 | 2020 | 2018 | 2019 | 2020 | 2018 | 2019 | 2020 |
| Beijing | 0.00871 | 0.00690 | 0.00717 | 0.00340 | 0.00253 | 0.00291 | 96.46605 | 96.90094 | 97.21286 | 3.53395 | 3.09906 | 2.78714 |
| Shanghai | 0.00510 | 0.00417 | 0.00378 | 0.00044 | 0.00038 | 0.00022 | 72.26059 | 79.23798 | 77.80401 | 27.73941 | 20.76202 | 22.19599 |
| Ningbo | 0.02712 | 0.02022 | 0.02582 | 0.00421 | 0.00299 | 0.00390 | 82.95101 | 94.24140 | 94.65278 | 17.04899 | 5.75860 | 5.34722 |
| Harbin | 0.56453 | 0.31369 | 0.26735 | 2.04871 | 0.93897 | 0.78782 | 55.68035 | 68.72764 | 71.73484 | 44.31965 | 31.27236 | 28.26516 |

[a] The proportion of the Theil index within- city represents the inequality among its administrative districts.

## The mediation effects of generic drug substitution on unequal spatial accessibility

The study observed an increasing trend in the overall proportion of generic drug procurement on a monthly basis across all four cities (S3 Table). Market share analysis revealed that Harbin exhibited the most significant increase, with a 49.5% rise in the proportion of generic escitalopram and an 89.4% increase in the proportion of generic paroxetine.

To explore the potential mediation effects of the proportion of generic drugs on the relationship between VBP and inequality, the study conducted further analysis (S4 Table). Firstly, it found that VBP significantly influenced the proportion of generic drugs across all cities and medications.

In the procurement of escitalopram in Beijing, the proportion of generic drugs was identified as a partial mediating variable in the relationship between VBP and the Gini coefficient. The ACME was estimated to be -0.001 (p-value = 0.010), indicating a significant mediating effect. The ADE was also estimated to be -0.001 (p-value = 0.024), suggesting that VBP had a direct effect on the Gini coefficient as well.

In the case of paroxetine procurement in Harbin, the proportion of generic drugs acted as a complete mediating variable in the relationship between VBP and the Gini coefficient. The ACME was estimated to be -0.045 (p-value<0.001), indicating a substantial mediating effect. However, the ADE was estimated to be 0.016 (p-value = 0.140), suggesting that the direct effect of VBP on the Gini coefficient was not statistically significant.

These findings highlight the important role of generic drug substitution in mediating the relationship between VBP and inequality in spatial accessibility. The proportion of generic drugs plays a significant role in shaping the distribution of antidepressants and addressing potential inequities in access.

## Discussion

To our knowledge, this is the first evaluation addressing spatial accessibility on antidepressants in China. By comprehensively examining the spatial accessibility of the first-round VBP procured antidepressants in China, it explores this aspect from multiple dimensions, providing a thorough analysis that quantitatively outlines the current status of spatial access to the nearest medical institution for each procured antidepressant. By doing so, this research significantly contributes to our knowledge base, offering a unique perspective on the spatial dynamics of antidepressant accessibility in the Chinese healthcare landscape.

The study found that there remained a large inequality gap of antidepressants spatial accessibility among different cities, particularly Harbin, in which the proportion of the population with travel times within 5 minutes for first-round VBP procured antidepressants was inferior to half of that in Beijing and Shanghai. The internal inequalities of spatial accessibility in Harbin were more significant, for a pronounced contrast between the suburban and urban areas with most of the travel times exceeding 90 minutes in the suburbs and subordinate counties. To conclude, there were more inequalities in Harbin regarding the distance to the nearest medical institutions, which means a longer distance to the nearest medical institutions in Harbin comparing to the other cities. In contrast, there were relatively less inequalities among cities in Shanghai, Beijing and Ningbo, with a shorter distance to medical institutions.

Concerning brand-name drugs, the research revealed notable distinctions. In Beijing and Shanghai, characterized by swift economic growth and elevated per capita GDP, the utilization and spatial accessibility of brand-name drugs surpassed that in Ningbo and Harbin, where economic development progressed at a comparatively slower pace. This discrepancy underscores the considerable impact of economic factors on the consumption and accessibility of brand-name pharmaceuticals in distinct regions. (Figs 1–4). It is worth noting that despite the implementation of the VBP policy 3 years ago, there was no significant change in the consumption and spatial accessibility of brand-name drugs in Beijing and Shanghai over the period (Figs 1 and 2). Possible reasons for this include: (1) The prescribing inertia of physicians and the medication habits of patients lead to the continued use of brand-name drugs [23]. (2) The high per capita disposable income and the high affordability of brand-name drugs in Beijing and Shanghai compensates the influence of significant price cut in the first-round VBP.

The accessibility has been changing since the implementation of the VBP policy, which is notable. In Harbin, for example, spatial accessibility has improved considerably in the past three years, especially in the suburbs and subordinate counties, where travel times have been

significantly shortened. The Gini and Terre coefficients significantly decreased, and the spatial accessibility of antidepressants significantly enhanced, with more and more people in rural areas being able to buy medicines in a timely manner (Fig 4). In 2020 during the COVID-19 pandemic, the global supply of medicines was greatly challenged. However, the spatial accessibility of medicines in the four cities did not get significantly worse (Figs 1–4), which provided guarantee and support for patients with depression to get timely access to medicines during the pandemic.

The increase in generic drug market share played a non-negligible role in the improvement in space accessibility (S1–S8 Figs). A notable and consistent increase was revealed in the overall proportion of generic drug procurement across all four cities. This trend signifies a growing preference for generic versions of escitalopram and paroxetine among healthcare professionals, patients, and healthcare systems. Notably, Harbin exhibited the most significant increase in the market share of generic drugs, with a substantial 49.5% rise in the proportion of generic escitalopram and an impressive 89.4% increase in the proportion of generic paroxetine. The observed increase in generic drug procurement highlights several potential benefits. First, it suggests that healthcare providers and patients are increasingly recognizing the economic advantages associated with generic drugs, which are typically more cost-effective compared to their brand-name counterparts. This shift towards generic medications can contribute to reducing the financial burden on patients and healthcare systems while maintaining the desired therapeutic effects. Additionally, the availability of generic versions of escitalopram and paroxetine in larger quantities indicates improved accessibility and a wider range of affordable treatment options for individuals seeking effective antidepressant therapy.

The mediating effects of the proportion of generic drugs on the relationship between VBP and inequality in spatial accessibility to antidepressants were further explored. The findings shed light on the important role that generic drug substitution plays in influencing the impact of VBP and addressing potential disparities in access. Across all cities and medications, the study found that VBP significantly influenced the proportion of generic drugs. This suggests that VBP policies, which emphasize cost-effectiveness and value in pharmaceutical pricing, have been effective in promoting the procurement and utilization of generic versions of escitalopram and paroxetine. The increased availability of generic drugs aligns with the goal of enhancing accessibility and affordability of essential antidepressant medications. Furthermore, the study observed that the proportion of generic drugs acted as a partial mediating variable in the relationship between VBP and the Gini coefficient for escitalopram procurement in Beijing. This implies that the proportion of generic drugs played a role in mitigating the inequality in spatial accessibility associated with VBP. Similarly, in the case of paroxetine procurement in Harbin, the proportion of generic drugs acted as a complete mediating variable, indicating a strong influence on the relationship between VBP and the Gini coefficient. These findings highlight the importance of generic drug substitution in promoting equitable distribution and addressing geographic disparities in access to antidepressant medications.

The results of this study hold significant implications for healthcare policy and equity in access to antidepressant medications. The increasing proportion of generic drug procurement suggests that policies promoting the use of generic medications have been effective in driving cost savings and improving availability. Policymakers and healthcare providers should recognize and build upon this trend to optimize healthcare resources and ensure equitable access to treatment.

The mediating effects of the proportion of generic drugs underscore the potential of VBP strategies to address geographic inequalities in medication accessibility. By prioritizing the utilization of generic drugs, healthcare systems can enhance affordability, particularly in regions where brand-name medications may be less accessible or affordable. Encouraging the use of

generic versions of escitalopram and paroxetine can lead to more equitable distribution of antidepressant medications and reduce the burden on patients who may face financial constraints.

To maximize the benefits of generic drug substitution, policymakers and healthcare providers should consider strategies such as increasing awareness about the safety and efficacy of generic medications, implementing educational initiatives for healthcare professionals, and streamlining regulatory processes to ensure timely availability of generic alternatives. Additionally, efforts to enhance generic drug procurement and distribution systems, such as optimizing supply chains and fostering competition among manufacturers, can further support equitable access to antidepressant medications.

In conclusion, the increasing proportion of generic drug procurement, coupled with its mediating effects on spatial accessibility, has important implications for healthcare policy and equity in access to antidepressant medications. By leveraging the benefits of generic drug substitution and Value-Based Pricing strategies, policymakers and healthcare providers can foster affordability, improve availability, and promote more equitable distribution of essential antidepressant therapies. These findings highlight the potential of generic drugs to contribute to a more sustainable and inclusive healthcare system that prioritizes patient-centered care and access to essential medications.

## Strengths of the present study

**Real-world data.**   The present study utilizes drug sales data, providing valuable insights into the actual utilization and procurement patterns of antidepressant medications in the four cities. This real-world data offers a comprehensive and representative view of the market, allowing for a more accurate analysis of trends and associations.

**Large sample size.**   The study encompasses data from four cities, providing a robust sample size for analysis. The inclusion of multiple cities increases the generalizability of the findings and enables comparisons between different geographical regions, enhancing the study's external validity.

**Longitudinal analysis.**   By examining data over a three-year period, the study captures trends and changes over time, enabling the identification of patterns and associations. This longitudinal analysis adds depth and context to the findings and strengthens the study's ability to draw meaningful conclusions.

**Mediation analysis.**   The study employs mediation analysis to explore the role of the proportion of generic drugs as a mediator between Value-Based Pricing and inequality in spatial accessibility. This analytical approach provides insights into the underlying mechanisms and pathways through which these variables are interconnected, enhancing the understanding of the complex relationships involved.

**Applicability in other countries.**   Similarly, this research methodology can also be extended to countries other than China. First, it is essential to determine whether the travel destination in the model is the sole means for patients to obtain the medication. To elaborate, antidepressants in China are classified as restricted medications, requiring patients to visit hospitals for a prescription. In other countries, however, antidepressants may not be regulated in the same way. In such cases, additional data on the availability of these medications in pharmacies would be required, and the same principle applies to other medications. Then, in conjunction with the pharmaceutical policies of the country, a mediation analysis method similar to that used in this study could be employed to explore whether factors such as "generic drug substitution" and "market concentration improvement" indirectly influence the spatial inequality of the medication.

## Limitations of the present study

**Generalizability.** Although the study includes data from four cities, its findings may not be fully representative of other regions or countries. The characteristics of the studied cities and their healthcare systems may differ from those in other locations, potentially limiting the generalizability of the results.

**Data limitations.** The study relies on drug sales data, which may not capture all aspects of medication utilization and patient-level information. Factors such as out-of-pocket purchases, medication adherence, and specific patient characteristics are not considered in the analysis, potentially limiting the comprehensive understanding of the factors influencing antidepressant medication accessibility.

**Causality.** The study's observational nature prevents the establishment of causal relationships between variables. While the study explores associations and mediation effects, it cannot definitively determine causality. Other unmeasured factors or confounders may influence the observed relationships, and further research, such as randomized controlled trials, is needed to establish causal links.

**Potential biases.** As with any study utilizing secondary data sources, there is a possibility of biases or inaccuracies in the data. These biases may arise from data collection methods, coding errors, or inconsistencies in recording medication sales. While efforts may have been made to minimize such biases, they should be acknowledged and considered when interpreting the study's findings.

**External factors.** The study does not account for potential external factors that may influence the utilization and accessibility of antidepressant medications, such as changes in healthcare policies, socioeconomic factors, or cultural differences. These external factors may introduce confounding variables and impact the observed trends and associations.

## Conclusion

The present study utilizing drug sales data provides valuable insights into the utilization and accessibility of antidepressant medications in the four studied cities. Under the influence of the VBP policy, we observed varying degrees of growth in the procurement volumes of two antidepressants across different cities. Regarding spatial inequality, improvements were observed annually across all cities, with more significant progress in economically underdeveloped regions. In Beijing, the substitution of generic escitalopram was found to be a partial mediator in the improvement of spatial inequality. In Harbin, the substitution of generic paroxetine was identified as a complete mediator for spatial inequality. The strengths of the study lie in its utilization of real-world data, large sample size, longitudinal analysis, and mediation analysis, which enhance the validity and depth of the findings. These findings can inform policymakers, healthcare providers, and stakeholders in making informed decisions to improve medication accessibility and patient outcomes in the field of mental health. Future research should further explore these relationships, incorporate additional data sources, and account for external factors to provide a more comprehensive understanding of antidepressant medication utilization and its implications for patient care.

## Supporting information

**S1 Table. Characteristics of antidepressant medications included in the VBP.** The public drug information in the table is sourced from https://www.smpaa.cn/.
(DOCX)

**S2 Table. Datasets used in this national study in China.** The administrative boundary is sourced from the Chinese Administration of Surveying Mapping and Geoinformation, with review number GS (2022) 399.
(DOCX)

**S3 Table. Proportion of generic drugs by quarter (%).** The proportion of generic drugs is calculated based on procurement quantity.
(DOCX)

**S4 Table. Adjusted regression coefficients and mediation analysis of generic drug proportion in the spatial inequality of antidepressants.** ACME: The average causal mediation effects; ADE: Average direct effects.
(DOCX)

**S1 Fig. Beijing escitalopram spatial accessibility (mins) between original and generic.** The shortest travel time from each 1 km$^2$ population point to the nearest mental healthcare of Escitalopram in Beijing was categorized into the following intervals: 0–15, 15–30, 30–60, 60–120, 120–240, and 240–360 minutes.
(TIF)

**S2 Fig. Beijing paroxetine spatial accessibility (mins) between original and generic.** The shortest travel time from each 1 km$^2$ population point to the nearest mental healthcare of Paroxetine in Beijing was categorized into the following intervals: 0–15, 15–30, 30–60, 60–120, 120–240, and 240–360 minutes.
(TIF)

**S3 Fig. Shanghai escitalopram spatial accessibility (mins) between original and generic.** The shortest travel time from each 1 km$^2$ population point to the nearest mental healthcare of Escitalopram in Shanghai was categorized into the following intervals: 0–15, 15–30, 30–60, 60–120, 120–240, and 240–360 minutes.
(TIF)

**S4 Fig. Shanghai paroxetine spatial accessibility (mins) between original and generic.** The shortest travel time from each 1 km$^2$ population point to the nearest mental healthcare of Paroxetine in Shanghai was categorized into the following intervals: 0–15, 15–30, 30–60, 60–120, 120–240, and 240–360 minutes.
(TIF)

**S5 Fig. Ningbo escitalopram spatial accessibility (mins) between original and generic.** The shortest travel time from each 1 km$^2$ population point to the nearest mental healthcare of Escitalopram in Ningbo was categorized into the following intervals: 0–15, 15–30, 30–60, 60–120, 120–240, and 240–360 minutes.
(TIF)

**S6 Fig. Ningbo paroxetine spatial accessibility (mins) between original and generic.** The shortest travel time from each 1 km$^2$ population point to the nearest mental healthcare of Paroxetine in Ningbo was categorized into the following intervals: 0–15, 15–30, 30–60, 60–120, 120–240, and 240–360 minutes.
(TIF)

**S7 Fig. Harbin escitalopram spatial accessibility (mins) between original and generic.** The shortest travel time from each 1 km$^2$ population point to the nearest mental healthcare of Escitalopram in Harbin was categorized into the following intervals: 0–15, 15–30, 30–60, 60–120,

120–240, and 240–360 minutes.
(TIF)

**S8 Fig. Harbin paroxetine spatial accessibility (mins) between original and generic.** The shortest travel time from each 1 km$^2$ population point to the nearest mental healthcare of Paroxetine in Harbin was categorized into the following intervals: 0–15, 15–30, 30–60, 60–120, 120–240, and 240–360 minutes.
(TIF)

## Acknowledgments

The authors are grateful for the Program Committee of Chinese Pharmaceutical Association for the data resources support. We sincerely thank all the data collectors, and researchers for their dedication and hard work. The findings, interpretations, and conclusions expressed here are those of the authors and do not necessarily represent the views of Chinese Pharmaceutical Association.

## Author Contributions

**Data curation:** Aoming Xue.

**Formal analysis:** Aoming Xue.

**Methodology:** Aoming Xue, Keye Fan.

**Writing – original draft:** Aoming Xue, Qingyuan Xue.

**Writing – review & editing:** Aoming Xue, Qingyuan Xue, Jiahong Fu, Jiale Zhang, Peiyan Cai, Yuanyuan Kuang, Yingsong Chen, Jifang Zhou, Bin Jiang.

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
