## [Decision Letter · Decision Letter 0]

16 Dec 2024

PONE-D-24-32343Quantifying the impacts of volume-based procurement policy on spatial accessibility of antidepressants via generic substitution: A four-city cohort study using drug sales dataPLOS ONE

Dear Dr. Zhou,

Thank you for submitting your manuscript to PLOS ONE. After careful consideration, we feel that it has merit but does not fully meet PLOS ONE’s publication criteria as it currently stands. Therefore, we invite you to submit a revised version of the manuscript that addresses the points raised during the review process.

We look forward to receiving your revised manuscript.

Kind regards,

Sheikh Arslan Sehgal, PhD

Academic Editor

PLOS ONE

6. We note that Figures 1, 2 ,3, 4, S1, S2, S3, S4, S5, S6, S7 and S8 in your submission contain [map/satellite] images which may be copyrighted. All PLOS content is published under the Creative Commons Attribution License (CC BY 4.0), which means that the manuscript, images, and Supporting Information files will be freely available online, and any third party is permitted to access, download, copy, distribute, and use these materials in any way, even commercially, with proper attribution. For these reasons, we cannot publish previously copyrighted maps or satellite images created using proprietary data, such as Google software (Google Maps, Street View, and Earth). For more information, see our copyright guidelines: http://journals.plos.org/plosone/s/licenses-and-copyright.

1. You may seek permission from the original copyright holder of Figures 1, 2 ,3, 4, S1, S2, S3, S4, S5, S6, S7 and S8 to publish the content specifically under the CC BY 4.0 license. 

Reviewers' comments:

Reviewer's Responses to Questions

**Comments to the Author**

1. Is the manuscript technically sound, and do the data support the conclusions?

Reviewer #1: Yes

Reviewer #2: Yes

2. Has the statistical analysis been performed appropriately and rigorously? 

Reviewer #1: Yes

Reviewer #2: Yes

3. Have the authors made all data underlying the findings in their manuscript fully available?

Reviewer #1: Yes

Reviewer #2: Yes

4. Is the manuscript presented in an intelligible fashion and written in standard English?

Reviewer #1: Yes

Reviewer #2: Yes

5. Review Comments to the Author

Reviewer #1: Thank you for the opportunity to review this well-written, interesting manuscript titled "Quantifying the impacts of volume-based procurement policy on spatial accessibility of antidepressants via generic substitution: A four-city cohort study using drug sales data". The manuscript is indeed well-written and the methodological choices are well justified. I would be happy to recommend publishing the manuscript.

Reviewer #2: In general, the manuscript appears to be technically sound. The study employs rigorous methods, including the least-cost-path algorithm, Gini coefficient, Theil index, Ordinary Least Squares (OLS), and mediation analysis to assess spatial accessibility and inequality of antidepressants. The data support the conclusions drawn, as the study found varied improvements in medication accessibility and inequality across cities, with more pronounced effects in economically less developed cities. Overall, the manuscript meets the standards for clarity, readability, and proper use of English.

1) The abstract should include the most significant results obtained during the research.

2) Line 165: "of the first round of the , paroxetine" – please check and revise this sentence for clarity and grammatical accuracy.

3) In the section "Data sources," appropriate references to websites or other sources should be provided where it is possible.

4) Line 189: It is recommended to include the formula in the main text of the manuscript.

5) The principles of the "volume-based procurement" strategy are not described but are valuable for a broad audience of readers.

6) Line 359: Should the abbreviation VBP here and afterward stand for "Value-Based Pricing" instead of "volume-based procurement"? Ensure consistency in the meanings of terms throughout the article.

7) Describe the applicability of the proposed methodology for conducting similar analyses in countries other than China.

8) In the Conclusion section, list the main results obtained in this study.

9) What are the main results expressed in numerical terms?

6. PLOS authors have the option to publish the peer review history of their article (what does this mean?). If published, this will include your full peer review and any attached files.

Reviewer #1: No

Reviewer #2: No

---

## [Author Response · Author response to Decision Letter 0]

10 Jan 2025

Dear Editor of PLOS ONE,

We appreciate your suggestions and reviewers’ kind comments on our submission entitled " Quantifying the impacts of volume-based procurement policy on spatial accessibility of antidepressants via generic substitution: A four-city cohort study using drug sales data (PONE-D-24-32343R1). This study is the first in China to conduct spatial accessibility research using antidepressant procurement data and has yielded some meaningful results. These results may provide some insights for both domestic and international readers. Once again, I sincerely appreciate the editor's patience in reviewing and handling my manuscript. We have carefully addressed these comments and please find below the point-to-point responses.

All my best, 

 Jifang Zhou, MD, PhD, MPH

Associate Professor

China Pharmaceutical University

Review Comments to the Author

Reviewer #1: Thank you for the opportunity to review this well-written, interesting manuscript titled "Quantifying the impacts of volume-based procurement policy on spatial accessibility of antidepressants via generic substitution: A four-city cohort study using drug sales data". The manuscript is indeed well-written and the methodological choices are well justified. I would be happy to recommend publishing the manuscript.

Response: Thank you very much for taking the time to review my manuscript. I am truly honored to receive your recognition. The VBP policy in China has directly impacted drug prices, procurement volumes, and other factors, but spatial accessibility is a novel perspective. Previous studies have primarily focused on the spatial accessibility of healthcare institutions at various levels, whereas our study is the first in China to use hospital procurement data to analyze spatial accessibility specifically in the field of antidepressant medications. We have also made several meaningful findings, and I believe these results will provide valuable insights for policymakers. I will continue to refine the content of the manuscript to meet the journal's requirements. Once again, thank you for your valuable review.

Reviewer #2: In general, the manuscript appears to be technically sound. The study employs rigorous methods, including the least-cost-path algorithm, Gini coefficient, Theil index, Ordinary Least Squares (OLS), and mediation analysis to assess spatial accessibility and inequality of antidepressants. The data support the conclusions drawn, as the study found varied improvements in medication accessibility and inequality across cities, with more pronounced effects in economically less developed cities. Overall, the manuscript meets the standards for clarity, readability, and proper use of English.

Response: Thank you very much for thoroughly reviewing my manuscript. I am also truly honored to receive your recognition of the strengths of the manuscript. Depression is highly prevalent in China, and patients need a prescription to purchase antidepressants from designated hospitals. Therefore, whether depression patients can access hospitals without spatial barriers is crucial. The VBP policy directly addresses drug prices and procurement volumes, but its impact on spatial accessibility and inequality remains unknown. Our study is the first in China to use hospital-level procurement data to analyze spatial accessibility in the specific context of antidepressants. We found that the substitution of generic drugs plays a significant role in improving spatial inequality. These findings are expected to provide valuable evidence for domestic policymakers and offer reference value for international research.

However, the manuscript does have some shortcomings. I greatly appreciate your constructive feedback and the valuable suggestions for improvement. I will address each of the issues you raised and make the necessary revisions, striving to meet your expectations.

(1) The abstract should include the most significant results obtained during the research.

Response: I completely agree with your suggestion, as the abstract of my manuscript is currently quite general and does not include the most significant results. To make the abstract clearer, I have revised it into a structured format and revised the results and conclusion sections. The revised version is as follows (Line 65 in Revised Manuscript with Track Changes):

Results: Under the influence of the VBP policy, we observed varying degrees of growth in the procurement volumes of two antidepressants across different cities (Escitalopram: Beijing 30.3%, Shanghai 26.2%, Ningbo 37.4%, Harbin 25.7%; Paroxetine: Beijing 28.2%, Shanghai 1.2%, Ningbo 50.2%, Harbin 590.5%). The increase in the procurement volumes of antidepressants across cities was primarily driven by generic drugs (Escitalopram: Beijing 159.8%, Shanghai 75.0%, Ningbo 146.4%, Harbin 146.3%; Paroxetine: Beijing 67.3%, Shanghai 4.9%, Ningbo 58.0%, Harbin 15,758.3%). In the results on spatial inequality, we observed annual improvements across all cities, with more pronounced progress in economically underdeveloped regions (Escitalopram: Gini in Harbin decreased by 10.6%; Paroxetine: Gini in Harbin decreased by 32.6%). In Beijing, the substitution of generic escitalopram was found to be a partial mediating factor in the improvement of spatial inequality (ACME = -0.00, p-value = 0.01; ADE = -0.00, p-value = 0.02). In Harbin, the substitution of generic paroxetine was identified as a complete mediating factor for spatial inequality (ACME = -0.04, p-value = 0.01; ADE = 0.01, p-value = 0.14). Conclusions: This study found that the spatial accessibility and inequality of antidepressant medications gradually improved under the influence of the VBP policy. These improvements can be partially attributed to the substitution of generic drugs.

In the revised version of the abstract, the main results of the study are included and specifically presented in numerical form, with a summary provided in the conclusion section. Thank you once again, and I hope the revised abstract meets your approval.

(2) Line 165: "of the first round of the, paroxetine" – please check and revise this sentence for clarity and grammatical accuracy.

Response: Thank you very much for your careful reading. This section indeed contains basic grammatical issues, which was my oversight. In the study area section, I intended to convey that we selected two antidepressant medications from the first batch included in the expanded VBP catalog as the subjects of this study. Without altering the original meaning, I corrected the grammatical issues in the sentence. The revised version is as follows (Line 198 in Revised Manuscript with Track Changes): In this study, we selected two depression medicines from the first round of the VBP: paroxetine and escitalopram.

(3) In the section "Data sources," appropriate references to websites or other sources should be provided where it is possible.

Response: I fully agree with your suggestion to include references to website information in the data sources section. This not only enhances the transparency of the research methodology but also makes the manuscript more rigorous. In this study, while the procurement data for medications was obtained from the Chinese Pharmaceutical Association, all other data sources are publicly available. This includes hospital coordinates, administrative boundaries, population information, and friction surface data. I have specified the URLs for each publicly available data source in the Data sources section (Line 208 in Revised Manuscript with Track Changes). For more detailed information on data sources, readers can refer to the Supporting information in S2 Table: Datasets used in this national study in China. Once again, thank you for suggesting this.

(4) Line 189: It is recommended to include the formula in the main text of the manuscript.

Response: Thank you very much for your reminder. The methodology section of my manuscript indeed lacks the necessary formulas to support the explanations. The Gini coefficient and other inequality indices were originally used to measure economic inequality in populations, but in this study, they are applied to assess spatial accessibility inequality within regions. Therefore, it is necessary to provide a specific explanation of how the parameters in the formulas are chosen, and I fully support your suggestion. Following your advice, I have included the formulas for the Gini coefficient and Theil index, along with the relevant explanations, in the methodology section of the main text (Line 223 in Revised Manuscript with Track Changes). I have also cited the appropriate references. This helps ensure that the calculation methods are clearly understood while maintaining the rigor of the writing.

(5) The principles of the "volume-based procurement" strategy are not described but are valuable for a broad audience of readers.

Response: Thank you very much for your suggestion from an objective perspective. I fully agree with adding a description of the principles of the VBP policy to help international readers gain a deeper understanding of the policy background. The VBP policy was piloted in China in 2018, following the principle of "price reduction through volume-based purchasing," and has yielded very positive results. To facilitate better understanding for international readers, I have used more vivid wording. The revised version is as follows (Line 132 in Revised Manuscript with Track Changes): The VBP policy follows the principle of "reducing prices through bulk purchasing," offering hospitals and patients more cost-effective medications.

(6) Line 359: Should the abbreviation VBP here and afterward stand for "Value-Based Pricing" instead of "volume-based procurement"? Ensure consistency in the meanings of terms throughout the article.

Response: Thank you very much for pointing out the inconsistency in the use of the VBP abbreviation. I completely agree with your perspective. "Volume-based procurement" is the correct expansion of VBP, and I have revised it accordingly (Line 411 in Revised Manuscript with Track Changes). "Volume-based procurement" better captures the essence of the VBP policy, as it reflects the concept of centralized bulk purchasing of medications. In contrast, "Value-Based Pricing" appears to refer more to the re-pricing of drugs. Similarly, I have also revised other instances of inconsistent usage of VBP throughout the manuscript to ensure such issues do not occur (Line 66, 93, 130, 198 in Revised Manuscript with Track Changes). Additionally, I ensured that the expanded form was used only when VBP first appeared in the abstract and main text, with subsequent mentions using the abbreviation. This aligns with the conventions of academic writing. Once again, I sincerely thank you for pointing out this issue in the manuscript.

(7) Describe the applicability of the proposed methodology for conducting similar analyses in countries other than China.

Response: I fully agree with your suggestion that a valuable study should provide reference value for other researchers in the field. As you mentioned, the research methodology used in this study is also applicable to similar research in other countries. Currently, studies on healthcare spatial accessibility in various countries focus on accessibility at the level of healthcare institutions, whereas research on the spatial accessibility of specific medications has not yet been developed. International scholars can refer to the methodology in this study for similar calculations of medication spatial accessibility, inequality, and related mediation analysis. 

Therefore, I have added a section titled "Applicability in Other Countries" in the "Strength of the Present Study" part of the manuscript for other researchers to reference. The added content is as follows (Line 451 in Revised Manuscript with Track Changes): Similarly, this research methodology can also be extended to countries other than China. First, it is essential to determine whether the travel destination in the model is the sole means for patients to obtain the medication. To elaborate, antidepressants in China are classified as restricted medications, requiring patients to visit hospitals for a prescription. In other countries, however, antidepressants may not be regulated in the same way. In such cases, additional data on the availability of these medications in pharmacies would be required, and the same principle applies to other medications. Then, in conjunction with the pharmaceutical policies of the country, a mediation analysis method similar to that used in this study could be employed to explore whether factors such as "generic drug substitution" and "market concentration improvement" indirectly influence the spatial inequality of the medication.

(8) In the Conclusion section, list the main results obtained in this study.

Response: Thank you very much for repeatedly pointing out the issue of unclear main results in my manuscript. I will revise the manuscript to address these issues. Similar to the changes made in the abstract, I have also incorporated main results in the conclusion section to support my findings. The main results of this study include three aspects: first, under the influence of the VBP policy, the procurement volume of antidepressants increased, and prices decreased. Second, the spatial accessibility and inequality of antidepressants improved annually. Finally, generic drugs were proven to be an effective mediating factor for this improvement. 

The added content is as follows (Line 488 in Revised Manuscript with Track Changes): Under the influence of the VBP policy, we observed varying degrees of growth in the procurement volumes of two antidepressants across different cities. Regarding spatial inequality, improvements were observed annually across all cities, with more significant progress in economically underdeveloped regions. In Beijing, the substitution of generic escitalopram was found to be a partial mediator in the improvement of spatial inequality. In Harbin, the substitution of generic paroxetine was identified as a complete mediator for spatial inequality.

(9) What are the main results expressed in numerical terms?

Response: Thank you very much for your valuable suggestion. I will make every effort to revise my manuscript accordingly. I understand that this feedback is similar to your earlier comments about the lack of clarity regarding the main research results in the abstract and conclusion. As a result, I have added quantified main results in both sections, with more detailed findings presented in the results section and tables (Line 76, 440 in Revised Manuscript with Track Changes). However, I only added numerical results in the abstract section, while in the conclusion section, I summarized the main findings of the article in a more concise manner. I hope these additions will help clarify the key findings of the study and provide readers with a clearer understanding of the research.

Your feedback has been extremely helpful, and I truly appreciate your valuable comments. Thank you once again for your thoughtful review.

Response: Thank you for the editorial comments and suggestions. I will carefully revise the manuscript to meet the journal's requirements. I have made the necessary revisions to the writing style based on the PLOS ONE guidelines, including adjustments to the font, font size, paragraph indentation, and other formatting requirements. I have also updated the file type and name as per the journal's specifications. If I have overlooked any style issues, I would greatly appreciate your guidance and feedback. Thank you very much.

(2) Please note that PLOS ONE has specific guidelines on code sharing for submissions in which author-generated code underpins the findings in the manuscript. In these cases, we expect all author-generated code to be made available without restrictions upon publication of the work.

Response: Thank you for pointing out the issue. I fully respect the guidelines on code sharing for submissions. The authors declare that the code used in t

---

## [Decision Letter · Decision Letter 1]

17 Jan 2025

Quantifying the impacts of volume-based procurement policy on spatial accessibility of antidepressants via generic substitution: A four-city cohort study using drug sales data

PONE-D-24-32343R1

Dear Dr. Zhou,

We’re pleased to inform you that your manuscript has been judged scientifically suitable for publication and will be formally accepted for publication once it meets all outstanding technical requirements.

Kind regards,

Sheikh Arslan Sehgal, PhD

Academic Editor

PLOS ONE

Additional Editor Comments (optional):

Reviewers' comments:

Reviewer's Responses to Questions

**Comments to the Author**

1. If the authors have adequately addressed your comments raised in a previous round of review and you feel that this manuscript is now acceptable for publication, you may indicate that here to bypass the “Comments to the Author” section, enter your conflict of interest statement in the “Confidential to Editor” section, and submit your "Accept" recommendation.

Reviewer #1: All comments have been addressed

Reviewer #2: All comments have been addressed

2. Is the manuscript technically sound, and do the data support the conclusions?

Reviewer #1: Yes

Reviewer #2: Yes

3. Has the statistical analysis been performed appropriately and rigorously? 

Reviewer #1: Yes

Reviewer #2: Yes

4. Have the authors made all data underlying the findings in their manuscript fully available?

Reviewer #1: Yes

Reviewer #2: Yes

5. Is the manuscript presented in an intelligible fashion and written in standard English?

Reviewer #1: Yes

Reviewer #2: Yes

6. Review Comments to the Author

Reviewer #1: (No Response)

Reviewer #2: Thank you for the opportunity to review the revised manuscript titled "Quantifying the impacts of volume-based procurement policy on spatial accessibility of antidepressants via generic substitution: A four-city cohort study using drug sales data". I appreciate the efforts of the authors in addressing the feedback provided during the review process. The authors have significantly improved the manuscript by addressing the concerns raised and incorporating the necessary changes to enhance its clarity and scientific rigor.

The methodological choices are well justified, and the study now provides a comprehensive and insightful contribution to its field. I am confident that the revised manuscript meets the high standards required for publication in your journal.

I am pleased to recommend this manuscript for publication and believe it will be of great interest to the readership. Thank you for considering my feedback and allowing me to contribute to this process.

7. PLOS authors have the option to publish the peer review history of their article (what does this mean?). If published, this will include your full peer review and any attached files.

Reviewer #1: No

Reviewer #2: **Yes: **Ruslan Z. Safarov

---

## [Editor Report · Acceptance letter]

24 Jan 2025

PONE-D-24-32343R1 

PLOS ONE

Dear Dr. Zhou, 

I'm pleased to inform you that your manuscript has been deemed suitable for publication in PLOS ONE. Congratulations! Your manuscript is now being handed over to our production team.

Kind regards, 

on behalf of

Dr Sheikh Arslan Sehgal 

Academic Editor

PLOS ONE